# Inhibitory Effect of *Canavalia gladiata* Extract on *Aggregatibacter actinomycetemcomitans* LPS Induced Nitric Oxide in Macrophages

**DOI:** 10.3390/nu17233764

**Published:** 2025-11-30

**Authors:** Eun-Sook Kim, Yun-Seong Lee, Jooyi Kang, Kang-Ju Kim, Yong-Ouk You

**Affiliations:** 1Institute of Wonkwang Dental Research, Wonkwang University, Iksan City 54538, Jeonbuk, Republic of Korea; bacteria-g@hanmail.net (E.-S.K.); iyoonseong@daum.net (Y.-S.L.); 2Department of Oral Biochemistry, School of Dentistry, Wonkwang University, 344-2, Shinyong-dong, Iksan City 54538, Jeonbuk, Republic of Korea; joo2qeen@hanmai.net; 3Department of Oral Microbiology & Immunol, School of Dentistry, Wonkwang University, 344-2, Shinyong-dong, Iksan City 54538, Jeonbuk, Republic of Korea

**Keywords:** *Canavalia gladiata*, *Aggregatibacter actinomycetemcomitans*, nitric oxide, concanavalin A, hemagglutination assay

## Abstract

**Objectives**: *Canavalia gladiata* (Jacq.) DC. (*C. gladiata*) has been used in traditional medicine to treat suppurative inflammatory conditions. Its antibacterial activity against oral pathogens and potential use as a non-alcoholic mouthwash have also been reported. This study aimed to elucidate the biological activity of Con A from *C. gladiata*, evaluate nitric oxide (NO) production induced by *Aggregatibacter actinomycetemcomitans* (*A. actinomycetemcomitans*), and as-sess its potential association with periodontal inflammation. **Methods**: In this study, a 0.5 M NaCl extract of *C. gladiata* (CGE_Na_) was prepared, and its protein content was quantified. The Concanavalin A equivalent (Con Aeq.) of CGE_Na_ was determined via a hemagglutination assay, and its effect on NO production was evaluated in RAW 264.7 macrophages stimulated with lipopolysaccharide (LPS-A.a.) derived from *A. actinomycetemcomitans*. LPS was extracted from six *A. actinomycetemcomitans* strains and used to induce inflammatory activation. **Result**: CGE_Na_ treatment significantly inhibited LPS-induced NO production at concentrations below 6.25 μg/mL without cytotoxic effects, suggesting an anti-inflammatory potential associated with lectin-like components. **Conclusions**: These results suggest that *C. gladiata* extract suppresses LPS-*A.a.*-mediated macrophage activation. Further studies are required to determine whether Con A specifically mediates this response and to evaluate its therapeutic relevance in the context of periodontal inflammation.

## 1. Introduction

*Canavalia gladiata* (Jacq.) DC. (*C. gladiata*), commonly known as sword bean, is native to tropical regions of East Asia and is currently cultivated in Korea [1]. The Jack bean contains non-protein amino acids such as Canavanine and Concanavalin A (Con A), as well as hemagglutinins found in many leguminous plants [2]. The Con A protein is a lectin, characterized by its ability to specifically bind to sugars on the cell surface and to be specifically inhibited by sugars [3]. Nevertheless, in Korea and Japan, the seeds and pods of *C. gladiata* have been used as a coffee substitute [4,5,6]. In folk medicine, *C. gladiata* has been employed to treat suppurative inflammatory conditions such as sinusitis, hemorrhoids, and boils, and its anticancer and antidiabetic activities have also been reported [7,8,9,10]. Moreover, *C. gladiata* exhibits strong antioxidant properties, and ethanol extracts have demonstrated anti-inflammatory effects in dextran sulfate sodium (DSS)-induced colitis models [11,12].

Recent studies further indicate notable antibacterial activities of *C. gladiata* against several pathogenic microorganisms. These include the methyl gallate–dependent inhibition of *Staphylococcus aureus* by methanolic extracts [13]; methanol or ethanol extract–mediated inhibition of foodborne pathogens such as *S. aureus*, *Vibrio parahaemolyticus*, and *Shigella sonnei* [14]; and the antibacterial activity of 80% ethanol extracts against *Vibrio vulnificus* and *V. cholerae*, with no inhibitory effect observed against *Streptococcus mutans* [15]. In addition, recent evidence—including that of the present study—demonstrates that sword bean extracts suppress the growth of *Porphyromonas gingivalis* and *Fusobacterium nucleatum*, as well as inhibit gingipain (Rgp and Kgp) activities, highlighting its potential as a natural agent for oral infection control and periodontal disease prevention [16]. Although the antimicrobial and anti-inflammatory properties of *C. gladiata* have been documented, most previous studies employed ethanol or methanol extraction. In contrast, the present study utilized a 0.5 M NaCl extraction to obtain Con A–containing preparations from *C. gladiata*, thereby enabling a focused evaluation of its lectin-associated bioactivity.

The Gram-negative oral bacterium *Aggregatibacter actinomycetemcomitans* (*A. actinomycetemcomitans*) plays a significant role in periodontal disease, particularly during the destructive phase involving rapid gingival and alveolar bone destruction [17,18]. The lipopolysaccharide of *A. actinomycetemcomitans* (LPS-A.a.) is a potent substance that induces the secretion of pro-inflammatory mediators from various cells, including epithelial cells, fibroblasts, and macrophages [19,20,21]. It has been reported that LPS-A.a. induces nitric oxide (NO) production in a macrophage cell line (J774) at concentrations 100 times lower than LPS derived from enteric bacteria [22]. Catalyzed by the three isoforms of NOS—neuronal NOS (nNOS), endothelial NOS (eNOS), and inducible NOS (iNOS)—L-arginine is metabolized into the gaseous molecule NO, which plays a crucial role in the cardiovascular, nervous, and immune systems [23]. Increased expression of iNOS has been reported in inflamed human periodontal tissues [24,25] and in experimental periodontitis in animal models [26,27], suggesting that oral bacteria such as *A. actinomycetemcomitans*, which induce periodontal tissue destruction [28], are responsible for the increased iNOS expression in periodontal tissues. This study aimed to characterize the biological activity of Con A derived from *C. gladiata* and to assess its potential association with periodontal inflammation by evaluating NO production induced by *A. actinomycetemcomitans*.

## 2. Methods

### 2.1. Extraction of Canavalia gladiata with 0.5 M NaCl (CGE_Na_)

Dried *C. gladiata* was purchased from Greenload Co. (Iksan, Republic of Korea) identified by Dr. Seung Il Jeong at the Jeonju Agrobio-materials Institute (Jeonju, Republic of Korea) and stored at 4 °C until use. Voucher specimen (no: 3-05-23) was stored in the Herbarium of the Department of Oral Biochemistry, School of Dentistry, at Wonkwang University. *C. gladiata* was pulverized using a grinder (Hanil Mixer, Gimpo, Republic of Korea), and 400 g of the powder was extracted in 500 mL of 0.5 M NaCl for 72 h at 4 °C. The mixture was then filtered through filter paper (Grade 5, 2.5 µm, Whatman International Ltd., Maidstone, UK) and centrifuged at 10,000 rpm for 20 min at 4 °C. After the resulting supernatant was filtered through a syringe filter, the protein concentration was determined using the Pierce™ 660nm Protein Assay Kit (Thermo scientific, Rockford, IL, USA) to standardize the protein content of the extract. The final concentration was adjusted with buffer, and the prepared extract was stored at −70 °C until use.

### 2.2. Chemicals and Reagents

3-(4,5-Dimethylthiazol-2-yl)-5-(3-carboxymethoxyphenyl)-2-(4-sulfophenyl)-2H-tetrazolium (MTS) was purchased from Promega (Madison, WI, USA). Griess Reagent (1% sulfanilamide in 5% H_3_PO_4_, 0.1% *N*-(1-naphthyl)-ethylenediamine dihydrochloride).

### 2.3. Bacterial Strains

*A. actinomycetemcomitans* ATCC 25923, 25922, and 33384 were purchased from the ATCC (Manassas, VA, USA). *A. actinomycetemcomitans* strains 1304, 1306, and CO114, isolated from Korean Collection of Oral Microbiology in Chosun University (Gwangju, Republic of Korea). The strains were cultured on Brain Heart Infusion (BHI) (Difco Laboratories, Detroit, MI, USA) at 37 °C in a 5% CO_2_ incubator.

### 2.4. Preparation of Erythrocytes

Erythrocytes were collected from rats. Whole blood samples were collected in centrifuge tubes containing an anticoagulant (sodium heparin), immediately washed five times with PBS (pH 7.2), and centrifuged at 1000 rpm for 30 min. The erythrocytes were resuspended in PBS and stored at 4 °C for experimental use. Animal welfare and experimental procedures were approved by the Institutional Review Board (IRB) of Wonkwang University (IRB No.: WKU25-60).

### 2.5. Hemagglutination Assay

Hemagglutination activity was assessed according to the technique described by Yunfei Zhang [29]. Briefly, the assay was performed using 96-well V-bottom microtiter plates (SPL, Pocheon, Republic of Korea). The CGE_Na_ (25 µg/mL) and Con A (1.67 µg/mL, hemagglutination standard) were prepared by serial two-fold dilutions in PBS, after which an equal volume (50 μL) of erythrocytes (1% *v*/*v*) in PBS was added to each well. The plates were incubated for 1 h at 4 °C or 37 °C, until the erythrocytes in the negative control wells had completely precipitated.

### 2.6. Extraction of A. actinomycetemcomitans LPS

*A. actinomycetemcomitans* was centrifuged at 12,000 rpm at 4 °C to completely remove the medium, and the pellet was washed three times with PBS. Lipopolysaccharide (LPS) from washed *A. actinomycetemcomitans* cells was extracted using a commercial LPS Extraction Kit (iNtRON Biotechnology, Seongnam, Republic of Korea) according to the manufacturer’s instructions. Briefly, 1 × 10^9^ *A. actinomycetemcomitans* cells were used for each extraction. The resulting LPS pellet was dissolved in 30 μL of 10 mM Tris-HCl buffer (pH 8.0) and stored at −20 °C until analysis. The working concentration of LPS-A.a. was set to 30 μg, based on the yield provided for *E. coli* in the manufacturer’s guidelines.

### 2.7. Cytotoxicity Assessment

To evaluate the cytotoxicity of CGE_Na_, RAW 264.7 cells (KCLB 40069, Korean Cell Line Bank, Seoul, Republic of Korea) were seeded in a 24-well plate at a density of 1.0 × 10^5^ cells/well and stabilized for 24 h. The cells were then treated with the extract at various concentrations (3.125, 6.25, 12.5, 25, and 50 µg/mL) and incubated for 48 h. After incubation, 3-(4,5-dimethylthiazol-2-yl)-5-(3-carboxymethoxyphenyl)-2-(4-sulfophenyl)-2H-tetrazolium (MTS, Promega, Fitchburg, WI, USA) reagent was added to the culture medium, and after 4 h, the absorbance was measured at 490 nm using an ELISA microplate reader.

### 2.8. Nitric Oxide Measurement

RAW 264.7 cells were seeded in a 24-well cell culture plate at a density of 1 × 10^5^ cells/mL per well and incubated for 24 h at 37 °C in a 5% CO_2_ incubator. To evaluate the post-treatment efficacy of CGE_Na_ on activated macrophages, RAW 264.7 cells were first stimulated with LPS-A.a. (2 µg/mL) for 1 h to initiate an inflammatory response, followed by treatment with CGE_Na_ for 48 h. The amount of NO produced was then measured by reading the absorbance at 570 nm using the Griess reagent [30].

### 2.9. Statistical Analysis

All experiments were performed in triplicate. Data were analyzed using the SPSS software (Statistical Package for the Social Sciences, version 19.5) and are presented as the mean ± standard deviation (SD). One-way analysis of variance (ANOVA) was conducted to evaluate differences among groups, followed by Tukey’s multiple-comparison test to determine statistical significance between individual means. Differences with a *p*-value of less than 0.05 were considered statistically significant.

## 3. Results

### 3.1. Analysis of Con A_eq_ in CGE_Na_

The Con A equivalent content (Con Aeq) of CGE_Na_ was determined using a hemagglutination-based functional assay, following previously established protocols [29,31]. This assay provides a relative index of lectin activity (Con Aeq) in comparison with standard Con A, and the values obtained reflect biofunctional equivalence rather than absolute quantification. In this study, the Con A content of CGE_Na_ was analyzed via a hemagglutination assay. The hemagglutination activity was compared to a Con A standard and converted to Con A_eq_ [31]. The calculation formula, based on a previous method [29], is presented below. According to the formula, 25 µg/mL of CGE_Na_ yielded 1.67 µg/mL of Con A_eq_ (Table 1, Figure 1). In this study, considering the cytotoxicity profile of CGE_Na_, a concentration of 6.25 μg/mL was selected for subsequent experiments. At this concentration, the extract was determined to contain 0.4175 μg/mL of Con A equivalents (Con Aeq).

The formula for calculating Con A_eq_ (Con A-equivalent) is as follows:C_Con A-eq_ = MAC_std_ × T_sample_ = 26.09375 ng/mL × 64 = 1.67 µg/mL$${\mathrm{MAC}}_{\mathrm{std}}=\frac{C_{0,std}}{T}=\frac{1670\;\mathrm{ng}/\mathrm{mL}}{64}=26.09375\;(\mathrm{ng}/\mathrm{mL})$$

C_0,std_: Initial concentration of standard (Con A = 1.67 µg/mL);

T: Titer of Con A (Titer = 64);

MAC_std_ (ng/mL): Minimum agglutination concentration of Con A;

T_sample_: Reciprocal of sample titer (Titer = 64).

All assays were performed in triplicate to ensure reproducibility.

### 3.2. Cytotoxicity Assessment of CGE_Na_

To evaluate the cytotoxicity of CGE_Na_ on RAW 264.7 macrophages, the cells were treated with various concentrations of CGE_Na_ (3.125, 6.25, 12.5, 25, and 50 µg/mL), and cell viability was measured using the MTS assay. As a result, cell viabilities were observed to be 119 ± 2.4, 107 ± 1.8, 88 ± 2.3, 63 ± 2.5, and 40 ± 2.1% relative to the control group, respectively, indicating no toxic effects at concentrations of 6.25 µg/mL or less. A minor elevation of NO levels was observed at the lowest CGE_Na_ concentration (3.125 µg/mL), possibly due to transient macrophage activation or assay variability. Thus, the slight increase at 3.125 µg/mL may reflect the lower end of such a biphasic response curve rather than a genuine pro-inflammatory effect (Figure 2).

### 3.3. Analysis of NO Production by Con A, an Active Component of C. gladiata, in RAW 264.7 Cells

To evaluate the effect of Con A, an active compound of *C. gladiata*, on NO production, RAW 264.7 cells were treated with LPS-A.a. (ATCC 25923; 2 µg/mL) or Con A (12.5 µg/mL). Cytotoxicity and NO generation were assessed. Unlike preventive assays where compounds are applied before LPS exposure, this study employed a post-treatment design to examine whether CGE_Na_ could attenuate ongoing inflammatory responses, reflecting a therapeutic relevance for established inflammation. Compared with the control (100 ± 2.1%), cell viability decreased markedly after LPS treatment (14.2 ± 3.8%), whereas Con A maintained normal viability (103 ± 2.3%). NO analysis showed that, relative to the LPS group (100 ± 2.6%), the control and Con A (12.5 µg/mL) treatments produced 32.1 ± 1.5% and 35.2 ± 1.2%, respectively (Figure 3).

### 3.4. Inhibitory Effect of CGE_Na_ on NO Production in RAW 264.7 Cells

To evaluate the inhibitory effect of CGE_Na_ on NO production, LPS (2 µg/mL) extracted from both *A. actinomycetemcomitans* ATCC reference strains and Korean clinical isolates were used to stimulate RAW 264.7 macrophages for 1 h. Subsequently, the cells were treated with CGE_Na_ at concentrations of 3.125 and 6.25 µg/mL and incubated for 48 h. The NO content in the culture supernatants was then quantified using the Griess reagent. As shown in Figure 4 and Figure 5, treatment with 3.125 µg/mL CGE_Na_ resulted in NO inhibition rates of 90.8 ± 2.1%, 91.9 ± 1.5%, and 91.0 ± 2.1% against LPS derived from ATCC strains 29522, 29523, and 33384, respectively. At 6.25 µg/mL, the inhibition rates were 73.2 ± 4.8%, 83.5 ± 2.4%, and 88.6 ± 2.7% for the same strains. Similarly, LPS extracted from the Korean isolates (strains 1304, 1306, and CO114) showed inhibition rates of 85.6 ± 2.2%, 82.6 ± 4.2%, and 86.4 ± 3.2%, respectively, when treated with 6.25 µg/mL CGE_Na_. These results clearly demonstrate that CGE_Na_ effectively sup-presses NO production induced by LPS from both reference and clinically isolated *A. actinomycetemcomitans* strains (Figure 4 and Figure 5).

The data in Table 2 were analyzed using one-way ANOVA followed by Tukey’s post hoc test (*p* < 0.05). Within each *A. actinomycetemcomitans* strain group, means labeled with different superscript letters (a–c) indicate statistically significant differences. Across all strains (ATCC 29522, 29523, 33384, 1304, 1306, and 114), treatment with LPS alone markedly increased NO production, which was normalized to 100 ± 3–5% (superscript c). In contrast, both the control and Con A (12.5 µg/mL) groups showed similarly low basal NO levels (approximately 35–39%), and these values were statistically identical (superscript a), indicating that Con A itself did not stimulate NO production. Notably, CGE_Na_ (6.25 µg/mL) significantly reduced LPS-induced NO production in all tested strains, showing intermediate values (73–89%) that were statistically distinct from both the LPS group (c) and the control/Con A groups (a) (superscript b). This pattern confirms that CGE_Na_ exerted a measurable inhibitory effect on NO production triggered by LPS-A.a. (Table 2).

## 4. Discussion

*A. actinomycetemcomitans* is one of the many bacterial species that constitute the oral microbiota [32]. Since the first reported case of endocarditis caused by *A. actinomycetemcomitans* in 1964 [33,34], this organism—although primarily an oral commensal—has been recognized as an opportunistic pathogen capable of causing a variety of systemic infections, including endocarditis, arthritis, meningitis, osteomyelitis, and pulmonary abscesses [35]. Periodontal disease is an inflammatory disorder of the tooth-supporting tissues that is primarily initiated by biofilm-forming oral bacteria such as *A. actinomycetemcomitans* [36]. The lipopolysaccharide (LPS) of *A. actinomycetemcomitans* acts as a potent virulence factor, triggering the release of inflammatory mediators that promote alveolar bone resorption [37]. Although antimicrobial agents have been investigated for their therapeutic efficacy in treating *A. actinomycetemcomitans*-associated periodontitis [38], the increasing prevalence of antibiotic resistance has highlighted the urgent need to develop novel therapeutic alternatives targeting this pathogen [39]. Con A is widely reported as a toxin in various literature, raising safety concerns [40]. However, while many beneficial medicinal effects of *C. gladiata* have been reported, its practical application is limited due to the toxicity of its Con A. This study estimated the Con A content of CGE_Na_ by calculating its Con A_eq_, which was derived from mouse erythrocyte agglutination activity. Because the structure and binding specificity of the agglutinin in *C. gladiata* are very similar to those of Con A, analyzing Con A content via Con A_eq_ is a convenient method, though it is not a precise quantitative analysis [31]. Therefore, this study concluded that hemagglutination-based estimation of Con Aeq provides a functional index of lectin activity rather than an exact quantification of protein concentration. This bioassay offers a convenient comparative measure, it does not substitute for more precise analytical methods such as HPLC, Western blot. So, the values reported here should be interpreted as relative estimates of lectin activity within CGE_Na_, not as absolute Con A concentrations. In this study, we observed the NO production effect of CGE_Na_ on LPS-A.a. induction by Con Aeq. At a CGE_Na_ concentration of 25 µg/mL, the Con Aeq was estimated to be 1.67 µg/mL, corresponding to approximately 6.7% of the extract. This level is markedly lower than concentrations reported to induce toxicity in vivo. For example, one study [41] demonstrated that Con A at 500 µg/mL triggered NO generation in the vitreous body of rabbit eyes, indicating that substantially higher doses are required to exert measurable toxic or pro-inflammatory effects. The concentrations employed in our experiments fall well below this threshold, supporting the safety of CGE_Na_ within the tested range. Nonetheless, given the known mitogenic and immunostimulatory properties of purified Con A, further systematic evaluation—including dose-dependent cytotoxicity, receptor-mediated responses, and in vivo tolerability—will be essential for determining whether Con A–containing extracts such as CGE_Na_ may be feasible for clinical translation. Furthermore, Con A made mouse peritoneal macrophages more susceptible to TLR ligand-induced NO production [42]. In peritoneal resident cells, concanavalin A-derived lectin induced NO production in mouse peritoneal cells in vitro and induced NO production both in vitro and in vivo [43]. Although most previous studies have reported that Con A induces NO production, oral administration of *C. gladiata* ethanol extract following DSS stimulation in a DSS-induced colitis mouse model resulted in the downregulation of NO signaling-related proteins. Con A has also been shown to inhibit NO synthesis by downregulating pro-inflammatory cytokines and suppressing NF-κB activation [12]. The influence of Con A on NO generation appears to depend on multiple factors, including its concentration, duration of exposure, and the specific cell type involved, which may explain the variability in NO responses observed across studies. Therefore, it can be inferred that Con A may elicit distinct NO responses depending on the experimental context and biological conditions.

In previous studies evaluating the inhibition of nitric oxide production induced by LPS-A.a., conventional anti-inflammatory reference compounds such as Nω-monomethyl-L-arginine (L-NAME) and polymyxin B have been used as positive controls. For instance, L-NAME has been reported to reduce NO production by approximately 80% at 500 μM under stimulation with 10 μg/mL LPS-A.a., whereas polymyxin B at 1 μg/mL and L-NAME at 1 μM demonstrated inhibitory effects of roughly 30% and 40%, respectively [44,45]. In comparison, CGE_Na_ at 6.25 μg/mL showed a moderate inhibitory effect on LPS-A.a.–induced NO production in our study, with suppression levels lower than those previously reported for these established positive controls. However, direct comparison should be interpreted with caution, as the LPS concentration used to induce inflammation varies substantially among studies and likely contributes to differences in the observed inhibitory efficacy. However, to further clarify these differences, it is important to note that CGE_Na_ contains a mixture of proteins, including Con A equivalents, whose inhibitory effects are likely mediated through indirect modulation of LPS-induced inflammatory signaling rather than direct enzymatic inhibition. In contrast, L-NAME is a competitive inhibitor of nitric oxide synthase (NOS) and therefore produces a more immediate and potent reduction in NO synthesis at the enzymatic level. Likewise, polymyxin B directly binds to the lipid A moiety of LPS, neutralizing endotoxin activity and preventing LPS–TLR4 engagement, a mechanism that is inherently more direct and robust than the multi-component, pathway-modulating activity of CGE_Na_. Finally, differences in molecular stability, bioavailability, and metabolic behavior between purified pharmacological agents and natural extracts may also contribute to the observed variability in inhibitory potency. Despite these mechanistic distinctions, the inhibitory activity demonstrated by CGE_Na_ suggests a biologically meaningful anti-inflammatory potential and supports further investigation under standardized and harmonized experimental conditions.

However, direct evaluation of iNOS mRNA or protein expression was not included. Therefore, the precise mechanism by which CGE_Na_ exerts its inhibitory effect—whether through the suppression of iNOS transcription, translation, or enzymatic activity—remains to be elucidated. Future studies will focus on quantifying iNOS and COX-2 expression and analyzing the involvement of the NF-κB and MAPK signaling pathways to clarify the molecular mechanisms underlying the anti-inflammatory effects of CGE_Na_. Therefore, the findings of this study provide a foundational basis for future investigations into the potential mechanisms by which *C. gladiata* may contribute to the amelioration of periodontal inflammation.

In summary, treatment with 6.25 µg/mL of CGE_Na_, equivalent to 0.419 µg/mL of Con A, exhibited no cytotoxicity and significantly reduced nitric oxide production in macrophages stimulated with LPS-A.a., indicating a non-toxic anti-inflammatory effect. Moreover, CGE_Na_ further attenuated NO generation directly induced by *A. actinomycetemcomitans*. These results suggest that Con A-containing components of CGE_Na_ immunomodulatory properties that may contribute to the suppression of periodontal inflammation. Collectively, this study provides preliminary evidence supporting the potential of *C. gladiata*-derived lectin-rich extracts as safe and effective candidates for the development of novel therapeutic approaches to manage *A. actinomycetemcomitans*-associated periodontal disease.

## 5. Conclusions

The Con A equivalent content of CGE_Na_ was estimated to establish a safe concentration range, and inhibition of NO production induced by LPS-A.a., a key virulence factor in periodontal disease, was observed. Significant NO suppression was confirmed across six *A. actinomycetemcomitans* strains, including three ATCC reference strains (29522, 29523, 33384) and three Korean clinical isolates (1304, 1306, CO114). Although the precise molecular mechanism remains to be clarified, these findings indicate that Con A-containing components of *C. gladiata* represent promising candidates for further anti-inflammatory and periodontal therapeutic research.

## Figures and Tables

**Figure 1 nutrients-17-03764-f001:**
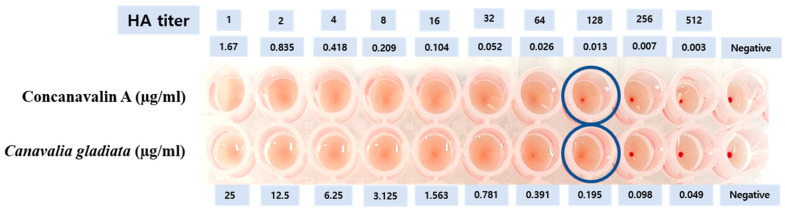
Determination of Con A_eq_ using the hemagglutination assay (HA). The wells indicated by circles represent the minimum concentration that exhibits hemagglutination.

**Figure 2 nutrients-17-03764-f002:**
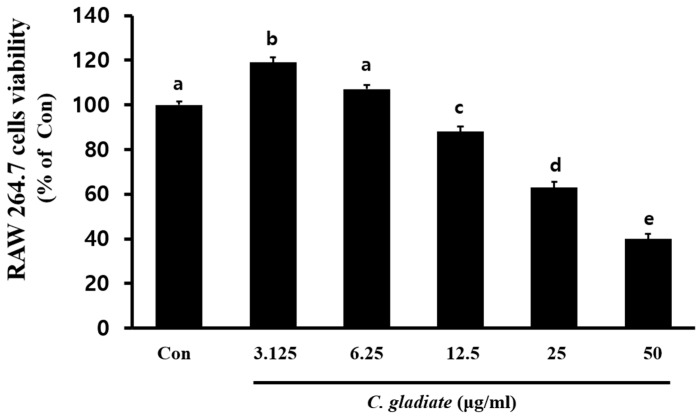
Cytotoxic effect of CGE_Na_ on RAW 264.7 cells. Cell viability was tested using the MTS assay. Data represent the results of three independent experiments and are expressed as the mean ± SD. Different superscript letters (a–e) denote statistically significant differences among treatment groups, as determined by one-way ANOVA followed by Tukey’s multiple comparison test (*p* < 0.05).

**Figure 3 nutrients-17-03764-f003:**
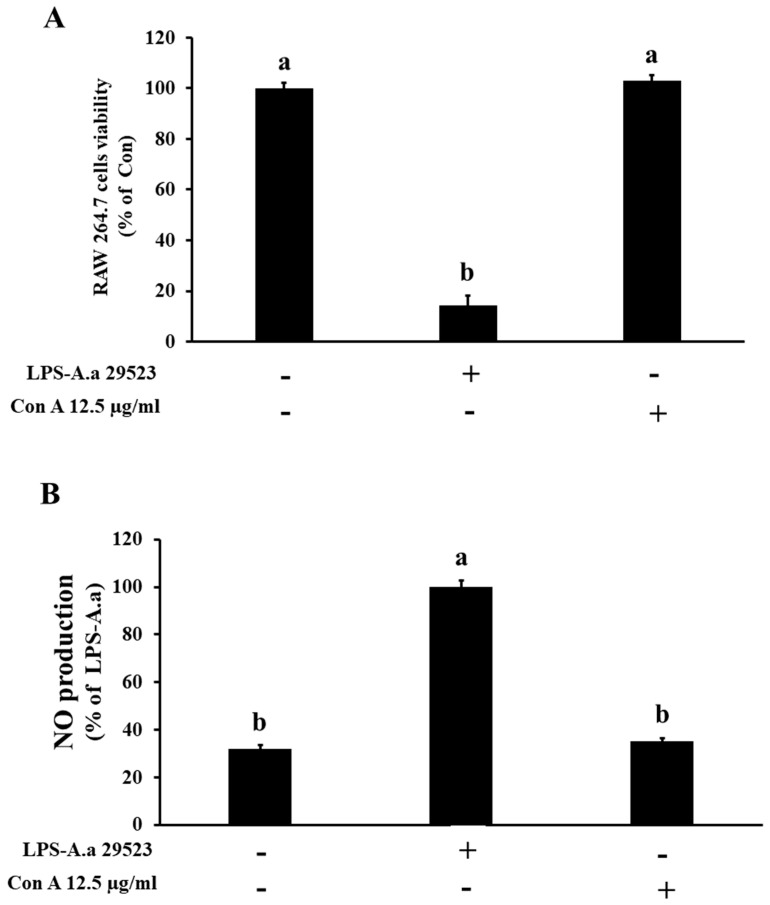
Inhibitory effects of Con A on NO production induced by LPS-A.a. in RAW 264.7 macrophages (ATCC strain 29523). (**A**) Cell viability measured by the MTS assay. (**B**) NO production measured using the Griess reagent. Cells were treated with 2 µg/mL LPS and 12.5 µg/mL Con A for 48 h. Data represent the mean ± SD (n = 3). Superscript letters (a, b) indicate statistically significant differences between groups according to one-way ANOVA followed by Tukey’s multiple comparison test (*p* < 0.05). +: Treatment, -: No treatment.

**Figure 4 nutrients-17-03764-f004:**
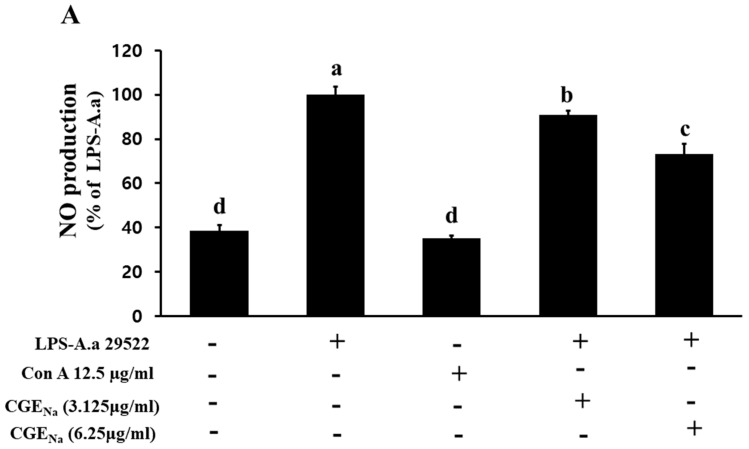
Inhibitory effects of CGE_Na_ on nitric oxide (NO) production induced by LPS-A.a. in RAW 264.7 macrophages (ATCC strains 29522; (**A**), 29523; (**B**), and 33384; (**C**)). Experimental conditions are shown in Figure 3. Cells were stimulated with 2 µg/mL LPS for 1 h and then treated with 3.125 or 6.25 µg/mL CGE_Na_ for 48 h. NO production in culture supernatants was quantified using Griess reagent. Data represent the mean ± SD (n = 3). Each superscript (a–d) indicates a statistically significant difference between treatment groups, as determined by one-way ANOVA followed by Tukey’s multiple comparison test (*p* < 0.05). +: Treatment, -: No treatment.

**Figure 5 nutrients-17-03764-f005:**
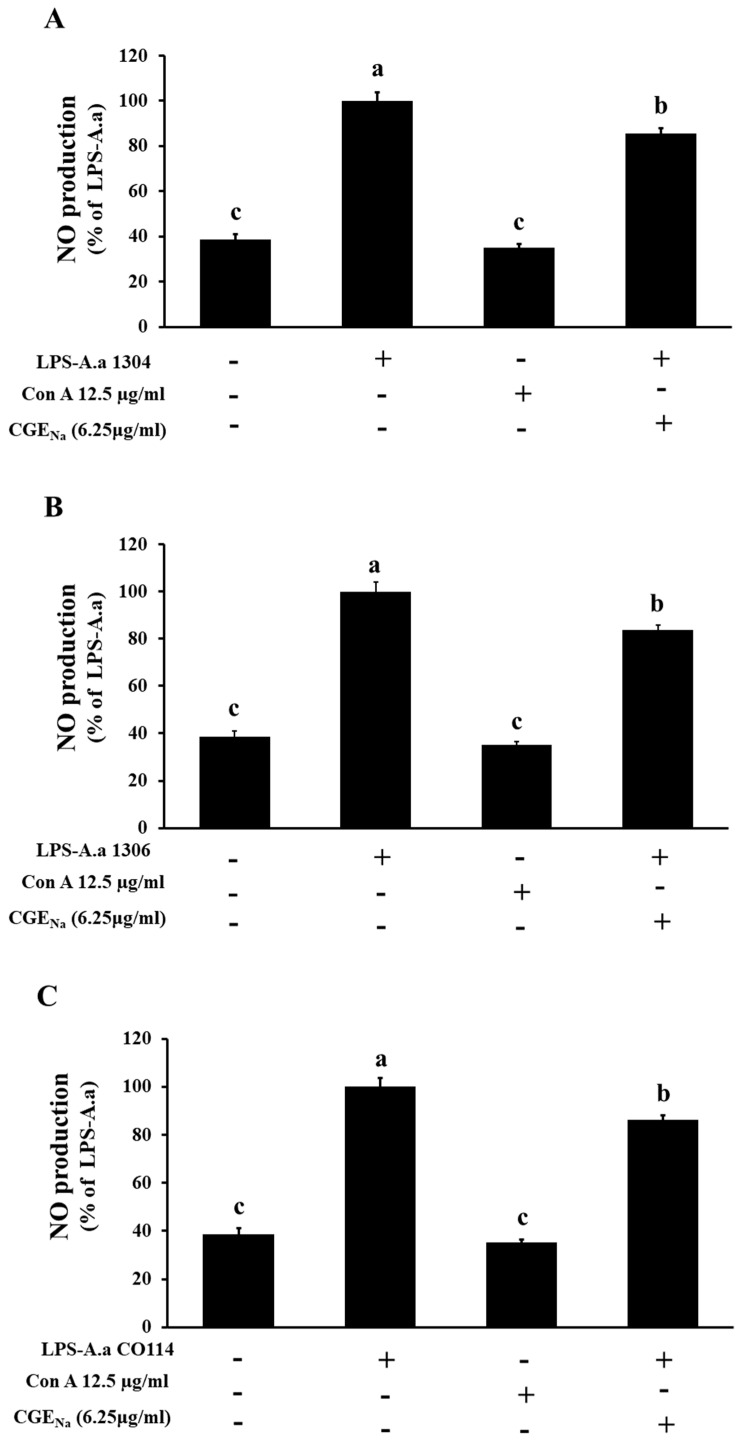
Effect of CGE_Na_ on nitric oxide (NO) production induced by LPS from Korean clinical isolates of *A. actinomycetemcomitans* (1304; (**A**), 1306; (**B**), and CO114; (**C**)). Experimental conditions are shown in Figure 4. The Korean isolate was treated only with CGE_Na_, a concentration of 6.25 µg/mL, which exhibits a pronounced NO inhibitory effect. Data represent the mean ± SD (n = 3). Each superscript (a–c) indicates a statistically significant difference between treatment groups, as determined by one-way ANOVA followed by Tukey’s multiple comparison test (*p* < 0.05). +: Treatment, -: No treatment.

**Table 1 nutrients-17-03764-t001:** Con A content of CGE_Na_.

Substance	The Concentration (µg/mL)	
CGE_Na_	25	6.25
Con A_eq_	1.67	0.4175

**Table 2 nutrients-17-03764-t002:** Inhibitory effect of CGE_Na_ (6.25 µg/mL) on NO production induced by LPS-A.a.

Group	LPS-A.a					
	29522	29523	33384	1304	1306	CO114
Con	38.6 ± 2.5 ^a^	38.6 ± 2.5 ^a^	38.6 ± 2.4 ^a^	38.6 ± 2.4 ^a^	38.6 ± 2.2 ^a^	38.6 ± 2.1 ^a^
LPS	100.0 ± 3.5 ^c^	100.0 ± 3.7 ^c^	100.1 ± 3.8 ^c^	99.9 ± 3.8 ^c^	100.0 ± 3.6 ^c^	100.0 ± 4.9 ^c^
Con A	35.1 ± 1.6 ^a^	35.1 ± 1.5 ^a^	35.1 ± 1.6 ^a^	35.1 ± 1.6 ^a^	35.2 ± 2.1 ^a^	35.1 ± 1.3 ^a^
CGE_Na_	73.2 ± 4.6 ^b^	83.5 ± 3.3 ^b^	88.6 ± 2.7 ^b^	85.6 ± 1.8 ^b^	83.5 ± 2.3 ^b^	86.4 ± 1.7 ^b^

Different superscript letters (a–c) within each column indicate statistically significant differences among treatments according to one-way ANOVA followed by Tukey’s multiple comparison test (*p* < 0.05).

## Data Availability

All data generated or analyzed during this study are available from the corresponding authors on reasonable request. The data are not publicly available due to legal and ethical reasons.

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
