# Peer review of "Inhibitory Effect of Canavalia gladiata Extract on Aggregatibacter actinomycetemcomitans LPS Induced Nitric Oxide in Macrophages"

_nutrients, 2025, doi:10.3390/nu17233764_

Round 1

Reviewer 1 Report

Comments and Suggestions for Authors

Major Comments

  1. The end of the Introduction should conclude with the importance, aims, and justification of the study, without results.
  2. Line 126: This is an unusual design. Conventionally, cells are treated with the test compound either before or at least simultaneously with an inflammatory stimulus to examine a prevention or inhibition mechanism. LPS pre-treatment followed by CGENa treatment tests post-treatment efficacy, not prevention.
  3. Lines 136-147: The Con A eq methodology based on hemagglutination is recognized as “not precise quantitative analysis” (line 206). For appropriate quantification of Con A, one should use ELISA, HPLC, or Western blot. The current methodology restricts the interpretation of dose–response.
  4. Lines 211-219: Authors quote contradictory reports on how Con A acts to affect the production of NO, but do not reconcile them. Clarify which mechanism applies and whether CGENa's effect results due to Con A or other components.
  5. Lacking mechanistic data: There is no result on iNOS protein/mRNA expression. Measuring NO alone is not sufficient. Show whether CGENa inhibits iNOS transcription, translation, or enzyme activity.
  6. Discussion repeats results, overemphasizes literature review and underemphasizes novel findings of the study. Focus on limitations, mechanistic implications and translational relevance

Minor Comments

  1. Figures 4-5: Bar graphs are missing error bars for control and LPS groups. All data should include standard deviations.
  2. Lines 224-225: “Con A content in 6.25 μg/mL CGENa = 0.419 μg/mL,” yet 12.5 μg/mL Con A showed no effect (lines 167-168). Explain this apparent discrepancy.
  3. Figure 2: At 3.125 μg/mL, viability = 119%, higher than control. Explain this stimulation or possible measurement error.
  4. Line 78 vs. line 87: Harmonize “CGENa” formatting. Line 111: centrifugation at 37°C is unusual (commonly 4°C). Add verification of LPS concentration and method of protein quantification.
  5. The title focuses on “nitric oxide production,” but there is limited mechanistic study beyond the measurement of NO. Suggest rewriting as: “Inhibitory Effect of Canavalia gladiata Extract on LPS-Induced Nitric Oxide in Macrophages.”
  6. No comparison with existing periodontal therapies is given. Address safety considerations concerning Con A toxicity (line 201) and discuss clinical translation.
Comments on the Quality of English Language
  1. The manuscript needs language refinement to make it clear and coherent. Several sentences need to be rewritten for readability, and overall stylistic polishing is needed to improve the flow.

Author Response

Title: Inhibitory Effect of Canavalia gladiata Extract on Aggregatibacter actinomycetemcomitans LPS Induced Nitric Oxide in Macrophages

 Summary

Thank you very much for taking the time to review this manuscript. Please find the detailed responses below and the corrections highlighted (red color) in the re-submitted files.

Reviewer 1

Comments 1: The end of the Introduction should conclude with the importance, aims, and justification of the study, without results.

Response 1: The last sentence of the introduction has been revised according to the reviewer's guidance (line 68).

Comments 2: Line 126: This is an unusual design. Conventionally, cells are treated with the test compound either before or at least simultaneously with an inflammatory stimulus to examine a prevention or inhibition mechanism. LPS pre-treatment followed by CGENa treatment tests post-treatment efficacy, not prevention.

Response 2: Xiaojing Gong et al. (2025); Yunhe Fu et al. (2014) Similar to other references, this study used LPS pretreatment as a common experimental model. Therefore, we revised the Methods and Results according to the reviewer's guidance (lines 125-128, 183-186).

Comments 3: Lines 136-147: The Con A eq methodology based on hemagglutination is recognized as “not precise quantitative analysis” (line 206). For appropriate quantification of Con A, one should use ELISA, HPLC, or Western blot. The current methodology restricts the interpretation of dose–response. 

Response 3: We are fully aware of the limitations of quantitative analysis of Con A content using hemagglutination reaction. The primary objective of this study was not to measure the absolute concentration of Con A, but rather to estimate the relative activity and functional equivalence of Con A in the CGENa extract for comparison and normalization purposes. Therefore, the Results and Discussion have been revised to reflect the reviewer's guidance (line 141-145, 266-272).

Comments 4: Lines 211-219: Authors quote contradictory reports on how Con A acts to affect the production of NO, but do not reconcile them. Clarify which mechanism applies and whether CGENa's effect results due to Con A or other components.

Response 4: The Discussion have been revised as follows, reflecting the reviewer's guidance     

            (line 292-296).

Comments 5: Lacking mechanistic data: There is no result on iNOS protein/mRNA expression. Measuring NO alone is not sufficient. Show whether CGENa inhibits iNOS transcription, translation, or enzyme activity.

Response 5: The Discussion has been rewritten based on the reviewer's guidance 

            (lines 311-319). 

Comments 6: Discussion repeats results, overemphasizes literature review and underemphasizes novel findings of the study. Focus on limitations, mechanistic implications and translational relevance.

Response 6: The Discussion has been rewritten based on the reviewer's guidance (lines 266-296).

Minor Comments

Comments 1: Figures 4-5: Bar graphs are missing error bars for the control and LPS groups. All data should include standard deviations.

Response 1:The submitted file shows error bars, likely due to the resolution. I will upload a high-resolution file.

Comments 2: Lines 224-225: “Con A content in 6.25 μg/mL CGENa = 0.419 μg/mL,” yet 12.5 μg/mL Con A showed no effect (lines 167-168). Explain this apparent discrepancy.

Response 2: At a concentration of 6.25 μg/mL, CGENa contained 0.419 μg/mL Con A equivalent and effectively inhibited NO production induced by LPS-A. actinomycetemcomitans stimulation (Fig. 3). However, treatment with 12.5 μg/mL Con A alone, in the absence of LPS-A. actinomycetemcomitans stimulation, resulted in a level of NO production comparable to that of the control group (Fig. 2). Therefore, these findings are not considered contradictory, as the experimental conditions and stimuli differ between the two assays.

Comments 3: Figure 2: At 3.125 μg/mL, viability = 119%, higher than control. Explain this stimulation or possible measurement error.

Response 3: The Result has been rewritten based on the reviewer's guidance 

             (lines 168-172).

Comments 4: Line 78 vs. line 87: Harmonize “CGENa” formatting. Line 111: centrifugation at 37°C is unusual (commonly 4°C). Add verification of LPS concentration and method of protein quantification.

Response 4: The Method has been rewritten based on the reviewer's guidance (lines 111). 

Comments 5: The title focuses on “nitric oxide production,” but there is limited mechanistic study beyond the measurement of NO. Suggest rewriting as: “Inhibitory Effect of Canavalia gladiata Extract on LPS-Induced Nitric Oxide in Macrophages.”

Response 5: The Title has been rewritten based on the reviewer's guidance (lines 2).

Comments 6: No comparison with existing periodontal therapies is given. Address safety considerations concerning Con A toxicity (line 201) and discuss clinical translation.

Response 6: The Discussion has been rewritten based on the reviewer's guidance 

             (comparison: lines 247-259, safety: lines 275-284)

Comments on the Quality of English Language

Comments: The manuscript needs language refinement to make it clear and coherent. Several sentences need to be rewritten for readability, and overall stylistic polishing is needed to improve the flow.

Response : Based on the reviewer's guidance, I requested 'Editage’ (English editing services) to revise the entire manuscript. Editing Certificate is attached.

Reviewer 2 Report

Comments and Suggestions for Authors

Insufficient description of extraction method innovation and comparative analysis with traditional methods.

The design of the concentration gradient is insufficient. Cytotoxicity tests only included 5 concentrations, lacking systematic investigation of lower concentrations. Study should include more low concentration points to establish complete dose-response relationship.

Lack of comparative analysis with other known anti-inflammatory drugs as positive control groups.

The time point Settings are single. Only 48-hour observation results, lacking dynamic time course data.

The details of the extraction method are insufficient. Lack of key parameters such as filter membrane pore size, extraction solvent volume. Provide comprehensive description of filtration conditions, centrifugation parameters in extraction process.

The method of Con Aeq calculation is unclear. Variable definitions in formulas are not clearly explained.

The bacterial culture conditions are missing. No detailed description of A. actinomycetemcomitans culture conditions.

The comparison between pure Con A and extracts‌ is inadequate. Lack of direct comparison of pure Con A under identical experimental conditions.

Statistical analysis is oversimplified. Only one-way ANOVA used, lacking multiple comparison corrections.

The dose-response relationship‌ is unclear. Absence of complete gradient data on CGENa concentration-dependent NO inhibition. Recommend supplementary studies on iNOS protein expression or related signaling pathways.

Current data supports the basic conclusion that "Canavalia gladiata extract inhibits A. actinomycetemcomitans LPS-induced NO production", but evidence supporting the extended conclusion that "Con A is a potential agent for ameliorating periodontal disease" remains insufficient.

Author Response

Title: Inhibitory Effect of Canavalia gladiata Extract on Aggregatibacter actinomycetemcomitans LPS Induced Nitric Oxide in Macrophages

 Summary

Thank you very much for taking the time to review this manuscript. Please find the detailed responses below and the corrections highlighted (red color) in the re-submitted files.

Comments 1: Insufficient description of extraction method innovation and comparative analysis with traditional methods.

Response 1: Previous studies on Canavalia gladiata have primarily utilized ethanol extraction to investigate the phytochemical constituents and their biological activities. In contrast, the present study focuses on the bioactive protein fraction, particularly Concanavalin A (Con A), which represents a distinct class of active components. To isolate this protein fraction, C. gladiata was extracted with a 0.45% NaCl solution rather than organic solvents, thereby optimizing conditions for protein solubility and functional preservation.

While several studies have reported Con A isolation and analysis using chromatographic and immunological techniques such as HPLC and Western blotting, more recent literature has introduced aptamer-based affinity extraction methods (reference). Although the current study does not quantify the precise Con A concentration, the Con A equivalents were determined based on hemagglutination bioactivity, providing a functional evaluation of the protein’s physiological relevance within the C. gladiata extract. This methodological distinction highlights the novelty of our extraction approach and its suitability for investigating protein-based bioactivities.

The Discussion has been rewritten based on the reviewer's guidance (lines 266-272)

Comments 2: The design of the concentration gradient is insufficient. Cytotoxicity tests only included 5 concentrations, lacking systematic investigation of lower concentrations. Study should include more low concentration points to establish complete dose-response relationship.

Response 2: In our preliminary experiments, we observed that nitric oxide (NO) inhibition at 3.125 µg/mL was lower than that observed at 6.25 µg/mL. Based on this finding, further below 3.125 µg/mL showed negligible NO suppression in preliminary assays, as these lower concentrations were unlikely to exhibit significant inhibitory effects. To address the reviewer’s concern, we have included the NO production data at 3.125 µg/mL for the A. actinomycetemcomitans ATCC reference strain in the revised manuscript. 

The Result has been rewritten based on the reviewer's guidance (Figure 4)

Comments 3: Lack of comparative analysis with other known anti-inflammatory drugs as positive control groups.

Response 3: The Discussion has been rewritten based on the reviewer's guidance (lines 297-311)

Comments 4: The time point Settings are single. Only 48-hour observation results, lacking dynamic time course data.

Response 4: In preliminary experiments, nitric oxide (NO) production in RAW 264.7 macrophages stimulated with A. actinomycetemcomitans LPS was found to be minimal at 24-hour points. Therefore, a 48-hour incubation period was selected to ensure sufficient accumulation of NO for reliable quantitative analysis. This time point was determined to be the most appropriate for observing measurable and biologically relevant changes in NO production under our experimental conditions.

Comments 5: The details of the extraction method are insufficient. Lack of key parameters such as filter membrane pore size, extraction solvent volume. Provide comprehensive description of filtration conditions, centrifugation parameters in extraction process.

Response 5: The Method has been rewritten based on the reviewer's guidance(lines 78-82)

Comments 6: The method of Con Aeq calculation is unclear. Variable definitions in formulas are not clearly explained.

Response 6: The Result has been rewritten based on the reviewer's guidance (lines 151-159)

Comments 7: The bacterial culture conditions are missing. No detailed description of A. actinomycetemcomitans culture conditions.

Response 7: The Method has been rewritten based on the reviewer's guidance (lines 93-94)

Comments 8: The comparison between pure Con A and extracts‌ is inadequate. Lack of direct comparison of pure Con A under identical experimental conditions.

Response 8: We would like to clarify that an analysis of pure Con A was performed and presented in Figure 3, and a direct comparative evaluation between pure Con A and the C. gladiata extract under the same experimental conditions is provided in Figure 4. This comparison demonstrates the relative inhibitory effects and biological activity of the extract relative to standard Con A, thus addressing the reviewer's concerns.

Comments 9: Statistical analysis is oversimplified. Only one-way ANOVA used, lacking multiple comparison corrections.

Response 9: The Result has been rewritten based on the reviewer's guidance (lines 245, Table 2)

Comments 10: The dose-response relationship‌ is unclear. Absence of complete gradient data on CGENa concentration-dependent NO inhibition. Recommend supplementary studies on iNOS protein expression or related signaling pathways.

Response 10: In the revised manuscript, we have included the data for 3.125 µg/mL CGENa using the A. actinomycetemcomitans ATCC reference strain. The results indicate that NO inhibition at 3.125 µg/mL was lower than that observed at 6.25 µg/mL, confirming a concentration-dependent trend within the effective range. Based on these findings, further analyses for the Korean clinical isolates were conducted only at the 6.25 µg/mL concentration, which demonstrated the most pronounced biological activity.

The Result has been rewritten based on the reviewer's guidance 

(dose-response relationshiplines 227-236, Figure 4iNOS protein expression: lines 311-319)

Comments 11: Current data supports the basic conclusion that "Canavalia gladiata extract inhibits A. actinomycetemcomitans LPS-induced NO production", but evidence supporting the extended conclusion that "Con A is a potential agent for ameliorating periodontal disease" remains insufficient.

Response 11: The Abstract has been rewritten based on the reviewer's guidance (lines 15-29)

Round 2

Reviewer 1 Report

Comments and Suggestions for Authors

The authors comprehensively addressed the reviewer's comments in the revised article. They modified the manuscript and figures, thoroughly answering all questions with detailed point-by-point responses and revisions.

Author Response

Thank you very much for taking the time to review this manuscript. 

Reviewer 2 Report

Comments and Suggestions for Authors

Supplement the specific quantitative results of the protein content of Canavalia gladiata extract (CGENa) instead of only mentioning "protein content quantification" to provide basic data for subsequent concentration-related analysis. If there is no quantification, the relevant expressions need to be modified.

Supplement the purity identification method and results of lipopolysaccharide (LPS-A.a) after extraction to ensure the effectiveness and consistency of the stimulant, and avoid affecting the experimental results due to differences in LPS purity.

Clearly explain the possible reasons for the differences in inhibitory effects between CGENa and positive control drugs (such as L-NAME, polymyxin B), such as action targets, drug metabolism characteristics, etc., to enhance the comprehensiveness of the discussion.

Correct the reference format errors (such as missing page numbers in some references and inconsistent DOI formats), and sort them out according to the reference specifications of the Nutrients journal.

Supplement the source information of the RAW 264.7 cell line used in the experiment (such as supplier, cell batch number) to enhance the reproducibility of the experiment.

Supplement the safe concentration range of CGENa (no cytotoxicity at 6.25μg/mL and below) in the abstract to make the core conclusion more complete.

Simplify the description of the traditional uses of Canavalia gladiata in the introduction, focus on anti-inflammatory and antibacterial activities related to periodontitis, and avoid redundant content.

Author Response

Summary

Thank you very much for taking the time to review this manuscript. Please find the detailed responses below and the corrections highlighted (red color) in the re-submitted files.

Reviewer 1

Comments 1: Supplement the specific quantitative results of the protein content of Canavalia gladiata extract (CGENa) instead of only mentioning "protein content quantification" to provide basic data for subsequent concentration-related analysis. If there is no quantification, the relevant expressions need to be modified.

Response 1: The Method has been revised as follows, reflecting the reviewer's guidance     

            (line 91-95).

Comments 2: Supplement the purity identification method and results of lipopolysaccharide (LPS-A.a) after extraction to ensure the effectiveness and consistency of the stimulant, and avoid affecting the experimental results due to differences in LPS purity.

Response 2: The Method has been revised as follows, reflecting the reviewer's guidance     

            (line 124-130).

Comments 3: Clearly explain the possible reasons for the differences in inhibitory effects between CGENa and positive control drugs (such as L-NAME, polymyxin B), such as action targets, drug metabolism characteristics, etc., to enhance the comprehensiveness of the discussion. 

Response 3: The Discussion have been revised as follows, reflecting the reviewer's guidance     

            (line 325-342).

Comments 4: Correct the reference format errors (such as missing page numbers in some references and inconsistent DOI formats), and sort them out according to the reference specifications of the Nutrients journal.

Response 4: The References have been revised to the Nutrients journal format, reflecting the reviewer's guidance (lines 383-480).

Comments 5: Supplement the source information of the RAW 264.7 cell line used in the experiment (such as supplier, cell batch number) to enhance the reproducibility of the experiment.

Response 5: The Method has been rewritten based on the reviewer's guidance 

            (lines 133-134). 

Comments 6: Supplement the safe concentration range of CGENa (no cytotoxicity at 6.25μg/mL and below) in the abstract to make the core conclusion more complete.

Response 6: The Abstract has been rewritten based on the reviewer's guidance (lines 24-25).

Comments 7: Simplify the description of the traditional uses of Canavalia gladiata in the introduction, focus on anti-inflammatory and antibacterial activities related to periodontitis, and avoid redundant content.

Response 7: The Introduction has been rewritten based on the reviewer's guidance (lines 42-62).
